# Organoids to Dissect Gastrointestinal Virus–Host Interactions: What Have We Learned?

**DOI:** 10.3390/v13060999

**Published:** 2021-05-27

**Authors:** Sue E. Crawford, Sasirekha Ramani, Sarah E. Blutt, Mary K. Estes

**Affiliations:** 1Department of Molecular Virology and Microbiology, Baylor College of Medicine, Houston, TX 77030, USA; crawford@bcm.edu (S.E.C.); ramani@bcm.edu (S.R.); sb691007@bcm.edu (S.E.B.); 2Department of Medicine, Baylor College of Medicine, Houston, TX 77030, USA

**Keywords:** enteric virus, human intestinal organoid, enteroid, gastrointestinal infection, diarrhea, host–virus interactions, Transwell^®^

## Abstract

Historically, knowledge of human host–enteric pathogen interactions has been elucidated from studies using cancer cells, animal models, clinical data, and occasionally, controlled human infection models. Although much has been learned from these studies, an understanding of the complex interactions between human viruses and the human intestinal epithelium was initially limited by the lack of nontransformed culture systems, which recapitulate the relevant heterogenous cell types that comprise the intestinal villus epithelium. New investigations using multicellular, physiologically active, organotypic cultures produced from intestinal stem cells isolated from biopsies or surgical specimens provide an exciting new avenue for understanding human specific pathogens and revealing previously unknown host–microbe interactions that affect replication and outcomes of human infections. Here, we summarize recent biologic discoveries using human intestinal organoids and human enteric viral pathogens.

## 1. Introduction

### 1.1. The Human Gastrointestinal Tract

The human gastrointestinal tract is a complex organ that functions as a barrier, absorbs nutrients, and responds to mcrobes. The luminal surface of the small intestine is populated with villi that contain a polarized epithelial layer of different cell types including enterocytes, enteroendocrine cells, tuft cells, goblet cells, and Paneth cells. These cells differentiate from progenitor Lgr5+ stem cells located at the base of intestinal crypts [1] (Figure 1a). Distinct regions of the intestine (duodenum, jejunum and ileum, proximal and distal colon) perform unique physiologic functions and demonstrate segment-specificity in terms of transport, protein expression, and interactions with pathogens. 

### 1.2. Human Intestinal Organoids

Human intestinal organoids (HIOs), also called enteroids, are cultures derived from stem cells in intestinal biopsies or surgical tissues (Figure 1b). First described in 2011 [2], success with producing these human cultures followed progress in the fields of developmental and stem cell biology primarily using mouse models [3,4]. The unique ability of HIOs to recapitulate basic organ structure and maintain the function and genetic diversity of the donor and specific tissue segment makes them exciting, advanced models for the study of human intestine–pathogen interactions. HIOs can be established from adult, pediatric, and fetal donors, genetically manipulated, and cryopreserved [5,6,7,8]. Once established, HIOs are easy to maintain indefinitely. However, the media used to support their growth is more expensive than most standard tissue culture media due to the need to include growth factors that promote stem cell proliferation. The media contains high concentrations of Wnt-3a, the WNT signal amplifier R-spondin, epidermal growth factor, and noggin, a secreted bone morphogenetic protein inhibitor (see [2,9] for more information on growth factors). Commercial media is available but expensive. Alternatively, cell lines that express WNT, R-spondin, and noggin to produce “conditioned media” are more cost-effective. Withdrawal of some of these growth factors redirects stem cells toward a differentiated cell state, resulting in cultures composed of all the major differentiated cell types that comprise the human small intestinal epithelium. Undifferentiated cultures contain stem cells, transit amplifying cells, immature enterocytes, and Paneth cells, whereas differentiated cultures contain enterocytes, enteroendocrine cells, goblet cells, tuft cells, and Paneth cells. 

Another stem cell approach to culture human “mini-intestines” uses induced pluripotent stem cells (iPSCs). iPSCs can undergo directed in vitro differentiation into a variety of organ cell types including human intestinal tissue [10,11]. Differentiation of iPSCs into organoids results in epithelial cells associated with mesenchyme; these cultures can be infected with human pathogens including rotavirus [12], norovirus [13,14], and SARS-CoV-2 [15]. However, these cultures take longer to generate than tissue-derived HIOs, they are limited to a finite number of passages, and in general, they retain a more fetal phenotype that makes them optimal for studying developmental biology but poses some limitations for studying host–pathogen interactions [16]. iPSC-derived intestinal organoids become more mature if transplanted into immunocompromised mice, a process that adds time and complexity to studies [17,18]. To our knowledge, transplanted iPSC-derived intestinal organoids have not been used to study enteric infection. This review mainly focuses on studies and new data obtained from tissue-derived HIO cultures.

Differentiated HIOs are often superior models compared to many previously used colon cancer-derived intestinal epithelial cell lines. First, cancer cell lines consist of a single epithelial cell population and thus lack the diversity of cell types that exist in the human intestine. In contrast, the presence of each intestinal cell type in differentiated HIOs allows investigation into intrinsic host receptors for susceptibility to infection, outcomes of infections, and intercellular crosstalk, including how one cell type influences neighboring cells or other cell types present in the intestinal epithelium. Another important limitation of cancer cell lines is that they frequently have altered signaling and metabolic pathways that can lead to misleading findings [19,20]. Critically for host–pathogen interaction studies, HIOs are infectable with many human-restricted pathogens including previously noncultivatable microbes such as human noroviruses, human astroviruses, *Salmonella Typhi*, and cryptosporidium [6,21,22,23]. Organoids also are revolutionizing the study of GI cell development and biology, allowing direct comparisons between humans and other species.

For many pathogens, much of our understanding of pathogen biology and disease pathogenesis comes from studies in animal models. However, there can be limitations of animal model research related to translation of animal data to humans. Viruses may exhibit strict species- and organ-specific tropisms, and some human viruses fail to infect or cause disease in animal models. This can be due to differences in the cellular receptors used by the viruses for entry into cells or physiologic differences that fail to reproduce human pathophysiology [6,24]. Additionally, significant differences exist between mice and humans in immune system development, activation, and response to challenge that can limit infection to animal but not human viruses [25,26].

### 1.3. Gastroenteritis Causing Viruses

Gastroenteritis causes 1.57 million deaths annually worldwide, with children under five years old representing one third of all who die [27]. Viral causes of gastroenteritis include human rotaviruses, noroviruses, sapoviruses, astroviruses, adenoviruses, and enteroviruses. Rotavirus infections were responsible for up to 500,000 deaths from diarrhea worldwide prior to vaccine implementation in 2006 [28]. While hospitalizations due to rotavirus-induced disease are markedly reduced since the introduction of vaccines, rotavirus infections still cause significant mortality with 128,000 deaths annually primarily in low- and middle-income countries where vaccine efficacy is low, highlighting the importance of continued study of this significant pathogen [28]. Concomitant with the decreased number of deaths associated with rotavirus following vaccination, human noroviruses, the leading cause of nonbacterial foodborne gastroenteritis worldwide, have replaced rotavirus as a principal cause of diarrheal illness in children where it has been examined [29,30,31]. Human astroviruses cause 2–14% of all acute nonbacterial gastroenteritis in children worldwide [32], and human sapoviruses cause 1–8% of milder outbreaks of gastroenteritis and some sporadic cases [33]. Human adenoviruses cause respiratory infections and conjunctivitis in addition to gastroenteritis, which is most often associated with infections with the enteric species F human adenoviruses serotypes Ad 40 and 41. Enteroviruses that include polioviruses, rhinoviruses, coxsackieviruses, and echoviruses are a significant cause of human infections worldwide and are shed in respiratory secretions and stools of infected individuals. Diarrhea is noted in infections by several respiratory viruses, but the pathophysiologic mechanisms of the intestinal infections are not known and may represent a poorly understood lung–gut axis of respiratory virus disease. Severe acute respiratory syndrome coronavirus (SARS-CoV-2), the causative virus of the current COVID-19 pandemic, infects the respiratory tract and gastrointestinal tract in some individuals and causes gastroenteritis symptoms including abdominal pain and diarrhea; the virus is shed in the stool of 20 to 50% of COVID-19 patients [34,35,36]. Other members of the *Coronaviridae* family, SARS-CoV and Middle East Respiratory Syndrome-CoV [36], closely related to SARS-CoV-2, have also been detected in the stool of infected individuals and cause gastroenteritis in addition to respiratory illness [37,38,39,40].

Despite the significant impact of gastroenteritis-causing viruses on human morbidity and mortality, fundamental knowledge of how these pathogens exploit the complex network of signaling pathways in human intestinal cells to establish a cellular environment conducive to drive faithful replication of their genomes remains limited. Progress in developing avenues to reduce the disease burden of these important human pathogens is emerging from detailed molecular, mechanistic, structural, and functional studies on each pathogen’s unique genes, proteins, and interactions with human hosts.

## 2. New Discoveries and Biology of Host–Microbe Interactions Revealed by HIO–Virus Studies

It is remarkable that although our knowledge of many components of the mucosal environment and of human pathogens is quite extensive, recent studies using ex vivo nontransformed HIO cultures are beginning to reveal previously unknown host–pathogen interactions that affect microbe replication and outcomes of human infections. HIOs provide an opportunity to study previously noncultivatable pathogens’ (human noroviruses and new strains of human astroviruses) host and tissue tropism as well as cell-specific responses to infection, identify new insights into disease mechanisms, and develop treatments and methods of viral inactivation (Table 1). New human gastrointestinal biology is also being revealed.

### 2.1. Human Viruses Infect More Intestinal Cell Types than Previously Recognized, and the State of Cell Differentiation Can Affect Infections

The technological advancement of ex vivo HIOs allows the identification of which cell types of the intestinal epithelium are infected by enteric viruses [6,21]. Enterocytes and enteroendocrine cells were identified as target cells infected by human rotaviruses and noroviruses in human intestinal biopsies and animal models [48,49,51,52]. In differentiated HIOs, enterocytes are also the predominant cell type infected by these viruses [5,6,50]. In addition, HIOs genetically engineered to expressed higher numbers of enteroendocrine cells confirmed human rotavirus infection of this cell type and revealed new biology [5,52,53]. Enteroendocrine cells are diverse specialized cells of the gastrointestinal tract that sense luminal contents and secrete multiple factors that regulate digestion, intestinal motility, and food intake. Rotavirus infection of enteroendocrine cells in HIOs induces the secretion of serotonin, the chemokine monocyte chemoattractant protein-1, glucose-dependent insulinotropic peptide, peptide YY, and ghrelin [53]. Stimulation of serotonin production from infected enteroendocrine cells is hypothesized to explain the predominance of vomiting in rotavirus infections in children [48], which is supported by a reduction of clinical symptoms with treatment with a serotonin receptor antagonist [54]. Whether secretion of serotonin and other molecules has additional functions remains to be determined. Whether human norovirus infects enteroendocrine cells in HIOs and stimulates serotonin production or other factors that mediate the vomiting observed in infected individuals requires further investigation.

Our knowledge of the tropism of adenoviruses, astroviruses, and enteroviruses has also been enhanced by experiments using HIOs. New information from studies of human adenoviruses in HIOs includes finding that human adenovirus strains associated with respiratory infections but with the capacity to cause gastrointestinal symptoms preferentially infect intestinal goblet cells. This contrasts with human enteric adenovirus strains that infect enterocytes [8]. In addition, enteric adenoviruses also are resistant to the human enteric alpha defensin HD5, while respiratory adenoviruses remain susceptible to the effects of this antimicrobial produced by Paneth cells [8,55]; these differences may reflect viral evolution/metamorphosis with enteric viruses evolving to overcome local cellular antiviral responses. Previously unrecognized differences in strain-specific tropisms for enterovirus infection of distinct intestinal epithelial cell types in HIOs include finding enterovirus 11 preferentially infects enterocytes and enteroendocrine cells, while enterovirus 71 replication occurs in goblet cells [46]. In contrast with other enteric viruses, human astroviruses infect not only mature enterocytes and goblet cells but also intestinal progenitor cells present in undifferentiated HIOs [21]. These studies demonstrate the power of HIOs to uncover aspects of enteric virus biology previously unattainable with standard cell lines. Understanding the consequences of these newly detected cell tropisms will help provide previously unappreciated insight on disease mechanisms. 

HIOs have been used to show that the novel severe acute respiratory syndrome coronavirus, SARS-CoV-2, can infect and replicate in intestinal cells [36,56,57,58]. Although COVID-19 patients have predominant respiratory clinical symptoms, they frequently exhibit gastrointestinal symptoms and shed SARS-CoV-2 virus in stool [59]. Stool detection initially raised questions of whether SARS-CoV-2 productively infects intestinal cells and whether fecal transmission occurs. Using intestinal organoids, several reports demonstrate SARS-CoV-2 can infect and replicate in undifferentiated proliferative cells, as well as in mature villous absorptive enterocytes in the small intestine (duodenum and ileum) and colonic large intestinal epithelial HIO cultures [36,57,58]. Infection and replication in undifferentiated proliferative cells were somewhat surprising, because mRNA expression of the SARS-CoV-2 receptor ACE2 is very low in these cells, while differentiated enterocytes express high levels of not only ACE2 but also the two mucosa-specific serine proteases TMPRSS2 and TMPRSS4 that facilitate SARS-CoV-2 spike protein fusogenic activity and promote virus entry into the host cell [58]. The similar infection rates of both proliferative cells and differentiated enterocytes suggest that low levels of ACE2 may be sufficient for viral entry [36]. A recent study of intestinal biopsies from COVID-19 positive patients detected patchy SARS-CoV-2 viral antigen in the duodenum and diffuse antigen detection in the ileum, confirming the studies in HIOs [35]. Intestinal susceptibility to infection has also been supported by studies that show individuals with proximal intestinal metaplasia (i.e., ectopic intestine-like cells in their esophagus) may have increased potential for virus to infect their epithelia [60]. This was supported by showing strong expression of both ACE2 and TMPRSS2 in intestinal-type metaplasia of the esophagus and stomach, unlike normal tissue, and that organoids from biopsies of Barrett’s esophagus faithfully reproduce the in vivo histopathologic characteristics of these patients and are susceptible to infection with a chimeric virus containing the SARS-CoV-2 spike protein [60]. Together, these studies indicate that replication in the human intestinal epithelium can occur, but whether virus detected in stool is infectious and contributes to fecal–oral transmission of the pandemic-causing SARS-CoV-2 remains unclear. This mimics previous results with the first SARS virus [61]. One study has indicated that the low pH of the stomach and the luminal fluid of the small and large intestines greatly attenuate the infectivity of SARS-CoV-2 that is present in swallowed oral secretions and refluxed respiratory sputum [58]. After a year of global infections, no epidemiologic studies clearly document fecal–oral transmission of infection, and only a few studies suggest virus in stool is infectious; instead, if the low pH of the gastric secretions as well as emulsifying conditions of the small and large intestines inactivate the virus, detected viral genomes in stool many not indicate infectious virus. Additional direct studies of the infectivity of stools containing different levels of virus are needed to definitively answer this question. 

### 2.2. Host Cell Polarity and Virus Infections

The ability to grow HIOs on permeable porous membrane inserts or plastic (i) allows the formation of a monolayer containing all the cell types present in the intestinal epithelium [62], (ii) facilitates study of the polarity by which viruses infect and are released from these epithelial cultures (Figure 1b and Table 1), and (iii) allows determination of whether infection disrupts barrier function, fundamental knowledge previously unattainable in human studies. Human rotavirus infects HIOs from all segments of the small intestine and the colon and reduces the barrier function of the epithelium as measured by loss in transepithelial resistance [5] (unpublished data). Human norovirus infects and is released from the apical surface of organoids from the small intestine, but not the colon from the same susceptible individuals [6,43]. The basis for this differential infection of human norovirus for small intestinal cells versus colonic cells remains unknown [6,43]. Infection with at least one pandemic strain of human norovirus causes a cytopathic effect of the monolayer [6]. SARS-CoV-2 preferentially (>1000-fold) infects the apical surface compared to the basolateral surface of HIO cells, which is consistent with the apical expression of the ACE2 receptor [58]. Enterovirus 71 binds and infects the apical surface of organoids, while enterovirus 11 exhibits an enhanced capacity to infect HIOs from the basolateral surface [47]. Enterovirus 11 is released from both the apical and basolateral compartments, although its release is skewed toward the basolateral compartment [47]. In contrast, enterovirus 71 is solely released from the apical compartment [47]. Enterovirus 71 does not alter epithelial barrier function but reduces the expression of goblet cell-derived mucins, suggesting that it alters goblet cell function [47]. In contrast, enterovirus 11 infection of HIOs induces significant damage of the epithelium with reorganization of tight junctions [46,47]. These results demonstrate that HIOs model early stages of enterovirus infections of the human intestinal epithelium and may allow study of how these viruses reach secondary sites of infection where more severe pathologies can develop. Determining the mechanism(s) by which viruses gain access to the apical or basolateral surface of intestinal cells and disrupt the epithelial barrier function may lead to the development of novel therapeutics to combat these pathogens.

### 2.3. Host Restrictions to Susceptibility to Viruses Are Recapitulated in HIOs

It is clear from many epidemiological studies that genetic factors can significantly affect susceptibility to mucosal pathogens in the intestine [63,64]. Several examples illustrate that HIOs are promising tools to understand host range restriction and how genetic polymorphisms affect susceptibility to infection. First, human rotavirus strains replicate well in HIOs compared to animal virus strains, providing the first ex vivo model of human rotavirus infection that recapitulates host range restriction [5]. Second, susceptibility of HIOs to human norovirus mimics the findings of epidemiological and human infection studies where susceptibility to infection is associated with secretor status determined by fucosyltransferase 2 gene (FUT2) expression [65,66]. Replication of the globally dominant, pandemic GII.4 strains is restricted to HIOs derived from secretor-positive persons who express a functional FUT2 gene, while non-pandemic GII.3 strains replicate in secretor-positive cultures and a subset of secretor-negative cultures. However, these initial correlative studies did not address whether FUT2 expression alone is critical for infection [6]. Using genetically manipulated isogenic HIO lines with or without FUT2 expression, Haga et al. have recently shown that FUT2 expression alone affects both human norovirus binding to the HIO cell surface and susceptibility to infection with several human norovirus strains (GII.4, GII.17, and GI.1) [67]. However, there are other host factors important for replication of at least one human norovirus (a GII.3) strain, as this strain still replicates in organoids where the FUT2 gene has been knocked out. These findings prove initial human norovirus binding to a molecule(s) glycosylated by FUT2 is critical for infection; in addition, they show that the human norovirus receptor is present in nonsecretor HIOs, as cultures become susceptible to infection by the sole expression of a functional FUT2 enzyme [67]. These results support the use of HIOs as a biological model to study the relationship between host genetics and susceptibility to viral infections.

### 2.4. Host Innate Responses to Infection

The intestinal epithelial layer forms a barrier between the lumen of the intestine and the host and is constantly exposed to commensal flora and pathogen challenges. The host cellular innate immune response plays a major role in the first line of defense against pathogen invasion by producing antiviral cytokines such as interferons (IFNs). Type I and type III IFNs are produced by epithelial cells and are essential for maintaining viral control and limiting viral infections. Type I IFN is widely expressed and involved in antiviral defenses generated by the rupture of epithelial barriers and/or viral dissemination and is associated with systemic inflammatory effects. Type III IFN has restricted expression (epithelial cells of respiratory airways, gastrointestinal tract, female reproductive system and placenta, and hepatocytes) and is involved in protection of epithelial surfaces to maintain viral control and immunity without activating and exacerbating an acute inflammatory response. The predominant innate intestinal epithelial response of human HIOs to several viral infections (human rotavirus, human norovirus, SARS-CoV-2, enterovirus 71) is a type III IFN response not previously detected in many infected cancer cell lines [21,41,44,45,46,47,56]. The direct effect of an intrinsic type III IFN response on viral replication is variable and is still poorly understood.

For rotavirus, infection of HIOs induces a predominant type III IFN transcriptional response, while infection of colon carcinoma-derived Caco-2 cells induces a strong type I IFN response at both the transcriptional and translational levels [41,68]. Unexpectedly, in the HIOs, only low levels of type III IFN protein are expressed early after infection, indicating a paradox of transcriptional and functional epithelial response to human rotavirus infection. Rotavirus antagonizes this endogenous IFN response at pre- and posttranscriptional levels with the result that rotavirus replication is not restricted [41]. The predominant transcriptional type III IFN response in HIOs is consistent with previous studies reporting that murine rotavirus-infected murine intestinal epithelial cells exhibit equal or greater fold inductions of type III IFN than type I IFN [69]. However, in rotavirus-infected mice, virus replication is reduced by type I IFN produced by immune cells. This finding is in agreement with the reduction of virus replication following exogenous addition of type I IFN in HIOs, suggesting that extraepithelial sources of type I IFN may be the critical IFN for limiting enteric virus replication in the human intestine [41,69].

Infection of HIOs by human norovirus strains, either a pandemic causing GII.4 virus or a GII.3 virus, induces a predominant type III IFN2/3 response [44,45] yet these strains exhibit differences in response to intrinsic cellular IFN pathways. While replication of both GII.3 and GII.4 human noroviruses is significantly reduced by exogenous treatment with type I or type III IFN, enhanced replication and spread of only GII.3, but not GII.4, virus is observed in HIOs genetically modified by knocking out the IFN signaling molecule STAT1 compared to replication in parental HIOs. These results indicate that cellular IFN responses restrict GII.3, but not GII.4, human norovirus replication and may provide one explanation for why GII.4 infections are more widespread [44]. Another newly recognized response to GII.4 infection of HIOs is in an increase in long noncoding RNAs, which are known to regulate gene expression [44]. Modulation of long noncoding RNAs has not been reported previously for gastrointestinal virus infections and is an important area for further study.

Enterovirus 71 replication induces and is restricted by type III IFN. Pharmacologic inhibition of IFN signaling with an ATP mimetic Janus kinase 1/2 inhibitor, ruxolitinib, increases enterovirus 71 infection [47], whereas enterovirus 11 infection, which also induces type III IFN, is more potently restricted by exogenous IFN-β treatment [47]. By contrast, human astroviruses induce a strong type I and type III IFN response in HIOs. Interestingly, human astroviruses infection of Caco-2 cells did not induce IFNs or interferon inducible genes, which again points to defects in innate immune sensing and/or signaling in this transformed cell line. The similar replication efficiency of enterovirus 11 in HIOs and Caco-2 cells, despite the robust induction of an antiviral response in HIOs, suggests that this virus has evolved mechanisms to evade this antiviral response in the intestinal epithelium. Furthermore, ruxolitinib treatment increased astrovirus replication in three HIO lines and was required in one HIO line, demonstrating that viral sensitivity to IFN signaling can be influenced by host genetic background and that there is a role for endogenous IFN responses in curtailing astrovirus replication [21]. Together, these findings of innate immune control of viral infections in HIOs is similar to what one would expect in vivo, where epithelial innate and adaptive immunity act synergistically to control and ultimately clear infections.

While SARS-CoV-2 replication is significantly impaired following exogenous treatment with either type I or type III IFN, the predominant IFN response induced by SARS-CoV-2 in HIOs is a type III IFN response, indicating that the type III IFNs play a critical role in controlling virus replication [56]. This contrasts with Caco-2 cells, which produce very large quantities of SARS-CoV-2 infectious virus and do not mount a robust innate immune response to infection [56]. A recent report of two large, independent cohorts of hospitalized patients in the United States and Europe revealed a significant reduction in mortality in COVID-19 patients with gastrointestinal symptoms compared with those without gastrointestinal symptoms, suggesting an “organ-specific” program of host response associated with a survival advantage [35]. Transcriptional profiling of intestinal tissue from patients revealed increased expression of anti-inflammatory gene products such as MUC2 mucin, a mucus-forming glycoprotein released by goblet cells that contributes to intestinal tolerance [70] and downregulation of a number of important proinflammatory gene products, including IFNG, IL1B, IKBKB, and STAT3B, which correlated with the lack of inflammatory infiltrates (monocytes, macrophages, and dendritic cells) in the intestinal epithelia of patients with gastrointestinal symptoms [35]. In addition, multiple proinflammatory cytokine/chemokine biomarkers associated with COVID-19 severity were found to be downregulated in COVID-19 patients compared with those without gastrointestinal symptoms [35,71,72,73]. To determine whether a similar response was observed in SARS-CoV-2-infected differentiated HIOs, the transcriptional profile of intestinal tissue from patients was compared to the transcriptional profile of SARS-CoV-2-infected differentiated HIOs (Figure 2). 

Many of the upregulated and downregulated genes identified in the intestinal epithelia of patients with gastrointestinal symptoms were detected in SARS-CoV-2-infected differentiated HIOs. The nuclear receptor RAR-related orphan receptor alpha (RORA) and IL1B, which are linked to Th17 cell differentiation and inflammatory bowel diseases, and the intestinal mucosa-specific IL-17C, shown to have pathogenic effects during SARS-CoV-2 infection, were depleted in both data sets. Antiviral response genes IFI44L, IFIT1, IFITM3, and IFI6 were upregulated in infected HIOs as well as in the intestinal epithelia of SARS-CoV-2 patients with gastrointestinal symptoms. Lastly, genes involved in immunomodulation including the metallothioneins, MT1E, MR1F, MT1H, MT1M, MT1X, and MT2A, were found to be upregulated in both datasets. These data suggest a previously unappreciated tissue-specific response to SARS-CoV-2 and provide rationale for continued study to determine whether this is a common mechanism for attenuation of not only intestinal SARS-CoV-2 but other enteric viruses. 

In general, the data with SARS-CoV-2 infection of intestinal organoids and the gastrointestinal tract may not be surprising given that the gastrointestinal viral infections are generally noninflammatory. Thus, rotavirus and norovirus both cause noninflammatory diarrheas compared with the inflammatory diarrheas induced by bacterial pathogens [41,44,74]. The ability of HIOs to recapitulate a similar epithelial-only response to viral and bacterial infections highlights the need for additional studies to determine the mechanisms by which viruses induce these different responses, which could lead to the identification of novel therapeutic targets and/or strategies to prevent or treat the pathogenesis and morbidity associated with enteric viral or bacterial infections. 

### 2.5. Dissection of Mechanisms of Pathogenesis

Diarrhea is a hallmark of most enteric viral infections. Under normal conditions, the gastrointestinal tract absorbs fluid and electrolytes, but during enteric virus infection, altered movement of ions through transporters or the lateral spaces between cells leads to diarrheal disease. HIOs are a unique physiologically responsive ex vivo model that can swell or exhibit luminal expansion in response to viral infection or exposure to known secretagogues such as cholera toxin [75]. Swelling represents a surrogate marker for fluid transport from the basolateral to the apical compartment, mimicking fluid secretion during diarrheal processes [75]. For example, HIOs rapidly swell in response to rotavirus infection or administration of the rotavirus enterotoxin [5]. The viral enterotoxin is a known novel calcium agonist that dysregulates host calcium (Ca^2+^) signaling pathways to increase cytosolic Ca^2+^, which is required for rotavirus replication. The viral enterotoxin drives these changes in Ca^2+^ homeostasis as both an endoplasmic reticulum-localized viroporin and a secreted enterotoxin [76,77,78,79]. These perturbations in host Ca^2+^ signaling activate autophagy, disrupt the cytoskeleton and tight junctions, and trigger fluid secretion pathways [77,80]. Recent studies have discovered a new component of rotavirus-induced signaling that includes paracrine signals that manifest as intercellular calcium waves observed in cell lines and HIOs; these waves are caused by the release of extracellular adenosine 5’-diphosphate that actives P2Y1 purinergic receptors on neighboring cells [81]. Blocking ADP signaling reduces rotavirus replication and inhibits rotavirus-induced serotonin release and epithelial fluid secretion in HIOs, causing less diarrhea in a mouse model [81]. These studies illustrate that HIOs can be useful to understand new aspects of the pathophysiology of infection, to test drugs to abrogate diarrhea, or to dissect the mechano- or pathophysiological processes that mediate diarrhea. Many more physiology and pathophysiology studies are expected. 

### 2.6. Neutralizing Antibodies, Therapeutics, Vaccines, and Viral Inactivation Evaluated in HIOs 

Studies with HIOs have allowed discovery of aspects of virus biology unattainable from or divergent with standard cell lines. Monoclonal antibodies specific for the cell binding domain (VP8*) of the rotavirus spike protein, previously identified as non-neutralizing antibodies when tested in a conventional monkey kidney cell assay, have been found to efficiently neutralize human rotavirus using HIOs [42]. Follow-up testing of other cell lines showed that similar results were observed in the intestinal human colon cancer cell line HT-29 but not other monkey kidney cell lines commonly used in rotavirus studies [42]. These results suggest that the identification of neutralizing epitopes on the rotavirus VP8* domain may have been underestimated, and the contribution of neutralizing antibodies induced in prior vaccine studies that evaluated human rotavirus neutralization responses may have been missed. Such results, in addition to basolateral infections in HIOs, suggest that intestinal cells may contain receptors that are not present on commonly used cell lines used for virus or bacterial studies.

With the ability to cultivate human noroviruses in HIOs, neutralization activity of human monoclonal antibodies and measurement of neutralizing antibody levels in infected/vaccinated persons can be evaluated [82,83,84,85]. Studies comparing neutralizing antibodies and a previously described surrogate of neutralization, histo-blood group antigen (HBGA)-blocking antibody, which was found to correlate with protection against clinical gastroenteritis in human studies, are now possible using HIOs. A first study comparing antibodies induced by administration of an investigational norovirus vaccine has shown a high correlation between these two (HBGA-blocking and neutralization) types of assays [83]. In addition, prior to replication of human noroviruses in HIOs, cultivatable surrogate viruses (murine norovirus, feline calicivirus, Tulane virus) were used to evaluate agents that inactivate these highly stable viruses. The recent use of HIOs to directly evaluate methods of human norovirus inactivation revealed that treatment with chlorine completely inactivates virus infectivity but alcohols, regardless of concentration or exposure time, only slightly reduce or do not completely inactive the tested human noroviruses [6,86]. These studies confirm the utility of HIOs as important tools to study mechanisms of neutralization and inactivation.

HIOs recapitulate in vivo observations of vaccine attenuation, with attenuated replication of the monovalent rotavirus Rotarix vaccine strain being seen in HIOs that is not observed in commonly used monkey kidney cell lines [5]. This result suggests that HIOs can be used to define mechanisms of vaccine attenuation. HIOs may also provide a platform for evaluating antivirals aimed at chronically infected immunocompromised patients [87]. FDA-approved drugs, natural compounds, and other drugs that block new pathways important for human norovirus or human rotavirus replication are beginning to be reported [81,88,89]. 

### 2.7. New Knowledge of Intestinal Biology Is Being Gained by Viral Infection Studies in HIOs

HIOs are a new model to uncover host components required for successful viral infection and obtain new knowledge of intestinal biology. One example is the discovery that the intestinal milieu can be critical for enteric viral infection. Thus, select human norovirus strains such as GII.3 require bile (or bile acids) to replicate in HIOs, and the replication of other strains is enhanced by these important factors found in the intestine [6,43,88]. 

Studies on the mechanism of action of the bile acid glycochenodeoxycholic acid (GCDCA) in jejunal HIOs found unexpectedly that this bile acid induces multiple cellular responses that promote human norovirus GII.3 replication in HIOs, including enhancement of (1) endosomal uptake, (2) endosomal acidification and subsequent activity of the endosomal/lysosomal enzyme acid sphingomyelinase (ASM), and (3) ceramide levels on the apical membrane. Inhibitors of endosomal acidification or ASM reduce GII.3 infection, and exogenous addition of ceramide alone permits infection. Furthermore, inhibition of lysosomal exocytosis of ASM, which is required for ceramide production at the apical surface, decreases GII.3 infection [88]. The membrane G protein-coupled receptor sphingosine-1-phosphate receptor 2 (S1PR2) is also required for replication [88]. Together, these results support a model where the bile acid-dependent GII.3 virus exploits rapid bile acid-mediated cellular endolysosomal dynamic changes and cellular ceramide to enter and replicate in jejunal HIOs [88]. This new information on human norovirus biology includes the identification of FDA-approved drugs (bile acid sequestrants (cholestyramine and colesevelam) and ASM inhibitors (amitriptyline)) that inhibit replication and may be useful to treat human norovirus infections providing they are tested and found to be efficacious in clinical trials.

New intestinal cell biology and differences in virus replication in HIO epithelial cells and classical tissue culture cells also are evident in studies on enteroviruses. The enterovirus genome contains a single long open reading frame, which is translated as a large polyprotein that is cleaved to produce the viral capsid and nonstructural proteins. A remarkable novel upstream open reading frame, uORF, that codes for the UP protein was described recently and found to be functional only in infected HIOs [7]. In infected differentiated HIOs, but not in other cancer cell lines, an UP-knockout affected echovirus 7 viral yield that was attenuated compared to wildtype virus was found. The membrane-associated UP protein appears to disrupt and facilitate virus release from virus enclosed membranes produced from the infected HIOs but not from other cell lines. UP protein expression was confirmed in poliovirus 1 and enterovirus 71 by western blot, and comparative genomic analyses have shown that the uORF is predominantly present in Enterovirus A, B, E, F, and G, as well as about half of C isolates. However, uORF is always absent for the respiratory tract infecting rhinoviruses, consistent with UP playing a specific role for enterovirus release in gastrointestinal tract epithelial cells. These findings overturn the longstanding dogma that enteroviruses use a single polyprotein gene expression strategy and may be important for new understanding of pathogenesis. 

HIOs are becoming an invaluable tool to study the functions of specific human intestinal epithelial cell types. For example, in cultures manipulated to express larger numbers of specialized enteroendocrine cells, gut hormone secretion is being dissected [53]. Another example is new insight being gained by the differentiation of microfold or membranous (M) cells previously studied mainly in mouse models. Peyer’s patches are specialized gut-associated lymphoid tissue in the small intestine that contain M cells located at the top of the Peyer’s patch dome. M cells are highly phagocytic and sample antigens in the intestinal lumen and transport the antigens to antigen-presenting cells lying underneath the follicle-associated epithelium. Lineage-tracing studies in mice demonstrated that M cell production is stimulated from intestinal stem cells by the cytokine receptor activator of the NF-κB ligand (RANKL), a type II member of the tumor necrosis factor superfamily [90]. Rouch et al. recapitulated that treatment of freshly isolated human intestinal crypts with recombinant RANKL stimulated M cell development similarly to the mouse studies [91]. These studies led to the development of a method to promote a subset of cells from ileal HIOs to differentiate into M cells by the addition of RANKL and TNF-α [92]. Human M cells can also be induced in HIOs using a combination of retinoic acid and lymphotoxin, and infection of these cultures showed that both reovirus and rotavirus transported across M cells are infectious, suggesting a mechanism for systemic dissemination of these viruses [93]. 

## 3. Summary and Current Questions

The remarkable discoveries highlighted above illustrate that studies in ex vivo human organoid cultures excel and can replace in vitro human-immortalized cultures, transformed cell lines, and primary cells for modeling human disease. These nontransformed epithelial cultures, which can represent human diversity and be expanded indefinitely, are facilitating the dissection of new mechanisms underlying biology and host–viral infections (Figure 3). In addition to the viruses described here, study of other gastroenteritis viruses such as hepatitis or emerging viruses in HIOs will permit future breakthroughs in understanding and modeling disease and developing interventions. Hepatitis E virus (HEV) can replicate in primary intestinal epithelial cells, which have a limited lifespan [94], so it could be advantageous to study HEV biology in intestinal organoid cultures that can be passaged indefinitely.

The discussed examples demonstrate why organoid models are gaining traction as the most physiologically relevant systems, short of infecting human volunteers, for studying interactions between commensals, pathogenic microbes, and humans. Many new questions remain to be answered (Table 2). A long-term goal is for organoid models to serve as a bridge between (or to replace) preclinical animal models and first-in-human volunteer studies or clinical trials. In this regard, HIOs are being evaluated for their ability to implement the Three Rs: replacement, reduction, and refinement of alternatives for animal experimentation. 

A next phase for using human organoids to fully understand infectious diseases requires adding culture complexity and well-integrated interdisciplinary research with biomedical engineers to develop technologies that are sufficiently simple for routine use in infectious disease laboratories and adequately robust for use in preclinical studies. With the expectation that human organoid models have enormous potential to fully understand human diseases, including human infectious diseases, limitations of current models need to be overcome. For example, the native intestine is more complex than the simple epithelium. Incorporation of components of the normal host microenvironment such as endothelial, immune, and neuronal cells and the microbiome, which can result in enhanced or reduced replication of different infecting microbes, will validate whether these cultures can fully recapitulate human mucosal tissues where most infections begin [62,95,96,97,98]. While it is unclear whether genetically matched tissue (mesenchyme, immune, and neuronal cells and the HIO) will be required to establish such cultures, human organoid cocultures offer the first controllable system to understand the mechanisms of immune or neuronal cells or microbial interactions and competition leading to enhanced or inhibitory outcomes. 

In addition, maintaining control over the self-organization processes, especially as cocultures are developed, will require engineered platforms that allow proper organization of the crypt–villus axis. Several platforms being evaluated model mechanical forces normally present in the intestine, including luminal flow and serosal blood flow (shear force) or peristaltic forces [99]. New data suggest that more mature intestinal architecture can be achieved by shaking organoids [100]. Other studies with an intestine-chip model to study *E. coli* heat-stable enterotoxin signaling suggest fluid flow, and not cyclic stretch, may be the critical mechanophysiologic cue needed for organoid growth and epithelial cell characteristics and infection studies [101]. Flow added to a millifluidic perfusion system increased viral titers [102]. Although more studies are certainly warranted, these data suggest flow may be the most important mechanical cue to evaluate in the next stage of organoid development. These advancements will require new creative biomedical research teams with expertise in virology, cell biology, physiology, immunology, and engineering. 

Other considerations need to be evaluated to replace the commercially supplied and expensive extracellular matrices (ECM) used to grow HIOs with in vitro generated ECM and to modulate and evaluate matric cues such as mechanical stiffness, degradability, and adhesive ligand presentation. The ECM regulates antiapoptotic pathways, cell differentiation, proliferation, and motility [103,104]. Corning^®^ Matrigel^®^ Matrix and BD Matrigel™ Basement Membrane Matrix are the most extensively used support matrices for culturing HIOs, but lot-to-lot variability, undefined composition, and cost hamper standardization of cultures. Collagen is an easily attainable connective tissue constituent; comparison of tissue derived HIOs grown in floating collagen I gels with Matrigel-grown organoids showed that on collagen, organoids aligned, fused, and formed macroscopic hollow tubes with a single enclosed lumen of budding crypt-like domains, more closely resembling the native architecture of the intestine [105]. New fully defined matrices based on hydrogels are being developed. Clearly, changing ECM constituents can increase control over organoid formation and/or organization. A research effort that shifts away from simply generating new organoid model systems toward improving new applications and establishing cocultures to successfully integrate missing cell lineages into new ex vivo systems to increase complexity and consistently determine their effects on infections may be fruitful. Finally, improving the ex vivo signaling environment and physical and/or mechanical environment using tunable matrices will allow for better spatiotemporal models to enhance our fundamental understanding of human biology and disease, especially infectious diseases. 

## Figures and Tables

**Figure 1 viruses-13-00999-f001:**
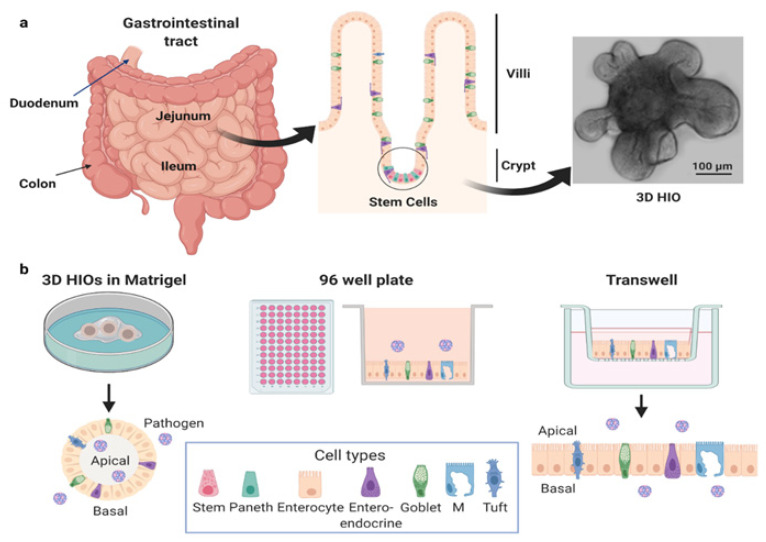
Human intestinal organoids. (**a**) Generation of HIOs from stem cells isolated from intestinal biopsies or tissue. (**b**) HIOs can be cultured as 3D HIOs in extracellular matrices such as Matrigel^®^ or plated, either as 2D cultures on 96 well plates or as polarized monolayers on Transwells^®^. HIOs can be plated on either on Matrigel^®^- or collagen-coated surfaces. In each of these formats, HIOs can be used as undifferentiated or differentiated 3D or 2D cultures. The apical and basolateral surfaces can be accessed by pathogens on Transwells^®^, while the apical surface of 3D HIOs is accessed by mechanical disruption. Created with BioRender.com (accessed on 26 May 2021).

**Figure 2 viruses-13-00999-f002:**
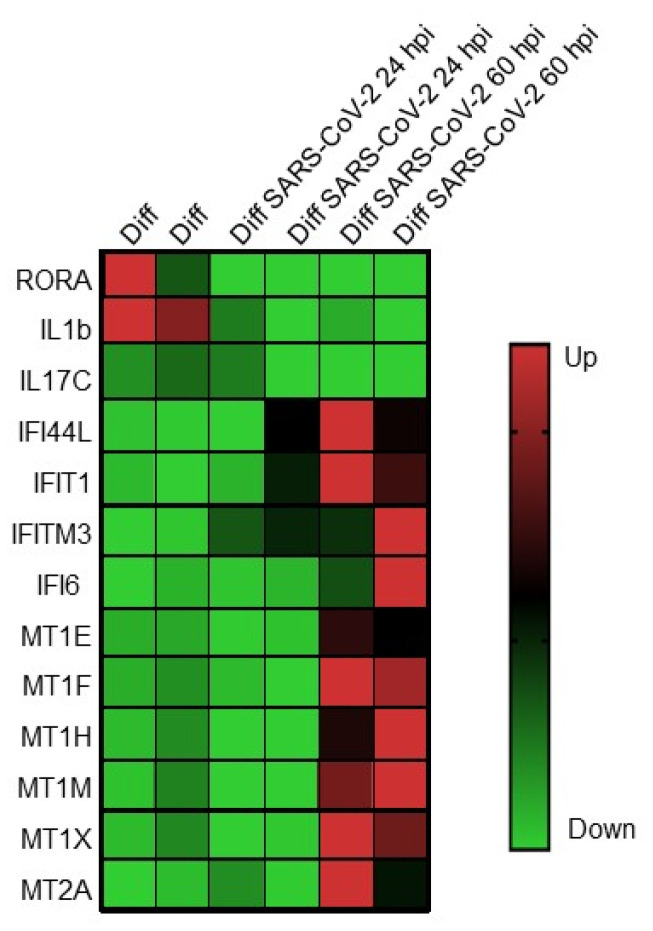
Expression of genes involved in innate immunity and immune modulation following infection of differentiated human intestinal organoids with SARS-CoV-2 coronavirus. Genes are listed on the left, and infection conditions are listed at the top. From [34].

**Figure 3 viruses-13-00999-f003:**
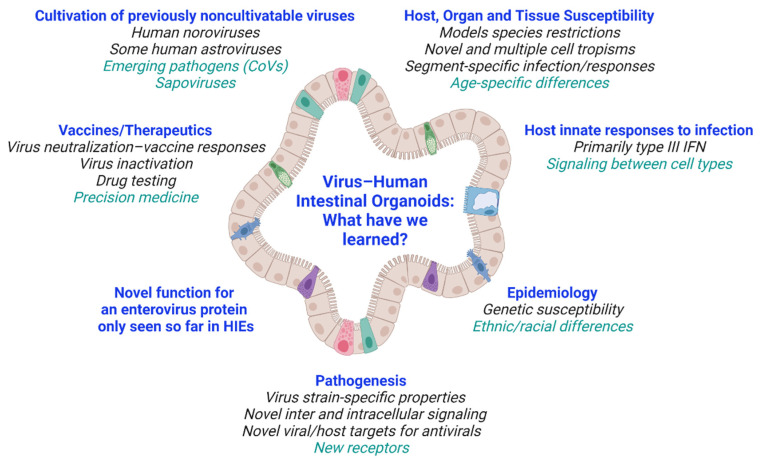
Overview of what we have learned from virus–HIO studies. Future areas of research are listed in green. Created with BioRender.com (accessed on 26 May 2021).

**Table 1 viruses-13-00999-t001:** Viral tropism, route of infection/release and interferon response to infection in HIOs.

Virus			Supports Virus Replication							
		Previously Noncultivatable	Undifferentiated/ Proliferative	Differentiated (Adult)	Susceptible Cell Type(s)	Route of Infection	Route of Release	IFN Response (Predominant)	Replication	References
Human Rotavirus		No	Poorly	D, J, I, C $	Enterocyte, Enteroendocrine	Baolateral > Apical	NT	Type III		[5,41,42]
Human Norovirus		Yes	No	D, J, I, not C	Enterocyte, Enteroendocrine	Apical **	NT	Type III	neutralized by monoclonal antibodies	[6,43,44,45]
Human Astrovirus		Some strains	D, J, I, C #	D, J **	Progenitor, Enterocyte, Goblet	Apical **	NT	Type I and Type III	inhibited by heat, 2CMC *; increased by ruxolitinib ***	[21]
Human Adenoviruses	Enteric	No	I **	I **	Enterocyte	Apical **	NT			
	Respiratory	No	I **	I **	Goblet	Apical **	NT		neutralized by enteric α-defensin HD5	[8]
Enteroviruses	Enterovirus 11	No	NT	Fetal **	Enterocyte, Enteroendocrine	Basolateral	Both but > Basolateral	Type III	increased by ruxolitinib	[46,47]
	Enterovirus 71	No	NT	Fetal **	Goblet	Apical	Apical	Type III		
SARS-CoV-2		No	Yes	D, I, C	Enterocyte	Apical 1000 fold > basolateral	NT	Type III	inhibited by imatinib, mycophenolic acid, quinacrine dihydrochloride ##	[36,48,49,50]

$ D (duodenum), J (jejunum), I (ileum), C (colon). # segments from adult and fetal patient donors. * Ribavirin and 2′-C-methylcytidine (2CMC), nucleoside analogues. ** only tested. *** ruxolitinib, an ATP mimetic janus-associated kinases. (JAKs) inhibitor that blocks STAT1 activation and inhibits ISG induction. ## imatinib, tyrosine kinase receptor inhibitor; mycophenolic acid, immunosuppressant, inhibitor of inosine monophosphate dehydrogenase; quinacrine dihydrochloride, suppresses NF-KB, activates p53 signaling and apoptosis. NT—Not tested.

**Table 2 viruses-13-00999-t002:** Current questions for HIO–enteric virus research.

Can epithelial cell responses to virus infection help explain pathogenesis?
Do goblet, enteroendocrine, or tuft cells have unique antiviral innate pathways to limit virus infection and/or replication? What mechanisms do viruses use to escape host innate responses?
Can novel viral receptor(s) be identified using polarized HIO monolayers in the transwell system?
Can viral interference, or bacterial–viral, and parasite–viral interactions, be modeled in HIOs?
Do all viruses use the same mechanisms to cross the intestinal epithelial barrier for systemic dissemination?
Can unique aspects of enteric virus replication, including dissecting slow viral kinetics, be elucidated in HIOs?
Will cell tropism for infection or epithelial cell responses differ in more advanced HIO cultures containing immune cells, innervation, or mesenchyme?

## Data Availability

Not applicable.

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
