# Peer review of "Organoids to Dissect Gastrointestinal Virus–Host Interactions: What Have We Learned?"

_viruses, 2021, doi:10.3390/v13060999_

Round 1
Reviewer 1 Report
Crawford et al have written an excellent, comprehensive review on the advances that have been made through the adoption of human intestinal organoids as models of authentic infection. For example, the differences in the interferon response (IFN) of continuous cell lines vs intestinal epithelial organoids is striking; similarly, interferon responses in mice, where in vivo mouse models are available, may not replicate the IFN response in human primary cells. This is a very timely review as we face pandemics such as COVID-19 caused by SARS-CoV2. Adherence to “classical” assays that rely on continuous human cancer cell lines, Vero cells, mice and other animal models is most likely contributing to our failure to effectively translate from bench to the clinic.
Minor points:
Line 26: should say “…luminal surface of the small intestine…” (the colon, large intestine, does not have villi)
Line 29: Barker et al., Nature 449:1003 (2007) should be cited at the end of the sentence (…intestinal crypts – cite here.). Isolation of Lgr5+ ISC and the discovery that these cells can self-organize to generate organoids would not have happened without the discovery of Lgr5 as an ISC marker and the mouse models generated.
Line3 90 typographical error – “these”
Lines 556/557 ECM not EMC
Table 1 suggest defining D, J, I, C in the table “legend” (assume that it is duodenum, jejunum, ileum, colon)
Author Response
We thank the reviewer for their comments that were helpful to improve this paper. Our responses to the reviewer’s comments and changes in the manuscript are outlined below.
Reviewer 1
Crawford et al have written an excellent, comprehensive review on the advances that have been made through the adoption of human intestinal organoids as models of authentic infection. For example, the differences in the interferon response (IFN) of continuous cell lines vs intestinal epithelial organoids is striking; similarly, interferon responses in mice, where in vivo mouse models are available, may not replicate the IFN response in human primary cells. This is a very timely review as we face pandemics such as COVID-19 caused by SARS-CoV2. Adherence to “classical” assays that rely on continuous human cancer cell lines, Vero cells, mice and other animal models is most likely contributing to our failure to effectively translate from bench to the clinic.
Minor points:
Line 26: should say “…luminal surface of the small intestine…” (the colon, large intestine, does not have villi)
“small” has been inserted into the text as requested.
Line 29: Barker et al., Nature 449:1003 (2007) should be cited at the end of the sentence (…intestinal crypts – cite here.). Isolation of Lgr5+ ISC and the discovery that these cells can self-organize to generate organoids would not have happened without the discovery of Lgr5 as an ISC marker and the mouse models generated.
Barker et al., Nature 449:1003 (2007) reference has been added.
Line3 90 typographical error – “these”
“these” has been corrected.
Lines 556/557 ECM not EMC
EMC has been changed to ECM.
Table 1 suggest defining D, J, I, C in the table “legend” (assume that it is duodenum, jejunum, ileum, colon)
D, J, I, C in Table 1 have been defined as duodenum, jejunum, ileum, colon.

Reviewer 2 Report
Crawford et al. presented an interesting review about the use of human intestinal organoids to study enteric viral pathogens. The review includes current knowledge about human intestinal organoids, the use of intestinal organoids to study host-microbe interactions , role of host cell polarity and viral infections, advantages of using human intestinal organoids to bypass the host restrictions, study of host innate immune response , the use of intestinal organoids to assess the neutralizing antibodies, therapeutics, vaccines and viral inactivation.
In general the review is interesting and easily flowed. I have some points that can improve the quality of the review
a) Please enumerate the requirements for the growth of crypt based stem cells. The authors only mentioned WNT protein. However, other components are required such as R-spondin, Noggin. In addition, other substances are required such as ROCK inhibitors, TGFB inhibitors, nicotinamide, NAC....etc. Please write shortly the role of these components in the growth of the organoids
b) Also the authors need to expand the basement matrix protein part required for organoid growth such as Matrigel, collagen
c) The authors should write a short paragraph on the limitations/ restrictions of the organoid model including the cost, need of trained skills, difficult to coculture several components together yo mimic the in vivo physiological gut .
d) The authors should differentiate in the review between 3D organoids (enteriod, stem cells, mostly undifferentiated) and 2D differentiate epithelium cells
e) In future applications, the organoids could be a promising tool to study other gastroenteritis viruses such as HAV, HEV, ..etc.
Author Response
We thank the reviewer for their comments that were helpful to improve this paper. Our responses to the reviewer’s comments and changes in the manuscript are outlined below.
Reviewer 2
Crawford et al. presented an interesting review about the use of human intestinal organoids to study enteric viral pathogens. The review includes current knowledge about human intestinal organoids, the use of intestinal organoids to study host-microbe interactions , role of host cell polarity and viral infections, advantages of using human intestinal organoids to bypass the host restrictions, study of host innate immune response , the use of intestinal organoids to assess the neutralizing antibodies, therapeutics, vaccines and viral inactivation.
In general the review is interesting and easily flowed. I have some points that can improve the quality of the review
- a) Please enumerate the requirements for the growth of crypt based stem cells. The authors only mentioned WNT protein. However, other components are required such as R-spondin, Noggin. In addition, other substances are required such as ROCK inhibitors, TGFB inhibitors, nicotinamide, NAC....etc. Please write shortly the role of these components in the growth of the organoids
Lines 42-50 address the critical components of the growth media and includes references for more information on other growth factors included in the media.
- b) Also the authors need to expand the basement matrix protein part required for organoid growth such as Matrigel, collagen
Lines 572-581 has been included to expand the requirement for ECM for organoid growth.
- c) The authors should write a short paragraph on the limitations/ restrictions of the organoid model including the cost, need of trained skills, difficult to coculture several components together yo mimic the in vivo physiological gut.
Cost and skills are addressed in lines 42-50.
Difficulties to coculture several components together to mimic the in vivo physiology of the gut is addressed in lines 546-554.
- d) The authors should differentiate in the review between 3D organoids (enteriod, stem cells, mostly undifferentiated) and 2D differentiate epithelium cells
Both the 3D and 2D can be either undifferentiated or differentiated based on whether the growth factors are withdrawn and allowed to differentiate as indicated in the legend of Figure 1. We and others have demonstrated infectivity of differentiated 3D cultures with enteric viruses. Table 1 shows whether undifferentiated or differentiated cultures are susceptible to the test viruses.
- e) In future applications, the organoids could be a promising tool to study other gastroenteritis viruses such as HAV, HEV, etc.
The following was added lines 518-521: In addition to the viruses described here, study of other gastroenteritis viruses such as hepatitis or emerging viruses in HIOs will permit future breakthroughs in understanding and modeling disease, and developing interventions.

Reviewer 3 Report
The review describes a subject on organoids from the gut for use in the establishment of the relationship between host-parasites considering viruses strains, mainly of intestinal origin. The manuscript is well written and worthy of publication. The use of animal´s tissues explants is particularly important for maintain the animal welfare, avoiding sacrifice the animals frequently. There is only a suggestion: the authors could replace "avenue" for "approaches" in the line 1
Author Response
We thank the reviewer for their comments that were helpful to improve this paper. Our responses to the reviewer’s comments and changes in the manuscript are outlined below.
Reviewer 3
The review describes a subject on organoids from the gut for use in the establishment of the relationship between host-parasites considering viruses strains, mainly of intestinal origin. The manuscript is well written and worthy of publication. The use of animal´s tissues explants is particularly important for maintain the animal welfare, avoiding sacrifice the animals frequently. There is only a suggestion: the authors could replace "avenue" for "approaches" in the line 1
The word "approaches" has been replaced with "avenue".

Round 2
Reviewer 2 Report
The authors adjusted the manuscript following my suggestions. I would like to thank the reviewer for this nice job.
One minor suggestion
In line 520: Please mention that up till today, the study of hepatitis viruses such as HEV is performed using epithelial cells isolated from organs such as kidney, GIT and endometrium, and this model is suitable for studying virus host interaction. However, this model has a limited passage number (PMID: 32824088, PMID: 31727684, PMID: 32316431). . Therefore, 3D organoid model can allow the growth and differentiation of cells for higher passage number.
Please include this part and cite these references (PMID: 32824088, PMID: 31727684, PMID: 32316431)
Author Response
The following sentence and the requested references are included in lines 520-523. For example, studies with hepatitis E viruses have been limited to infection of primary epithelial cells isolated from the kidney, gastrointestinal tract and endometrium and passage of these primary cells is limited [94-96]. The ability to indefinitely cultivate HIOs overcomes this limitation.